# Validated Reversed-Phase Liquid Chromatographic Method with Gradient Elution for Simultaneous Determination of the Antiviral Agents: Sofosbuvir, Ledipasvir, Daclatasvir, and Simeprevir in Their Dosage Forms

**DOI:** 10.3390/molecules25204611

**Published:** 2020-10-10

**Authors:** Essam Ezzeldin, Nisreen F. Abo-Talib, Marwa H. Tammam, Yousif A. Asiri, Abd El-Galil E. Amr, Abdulrahman A. Almehizia

**Affiliations:** 1Pharmaceutical Chemistry Department, College of Pharmacy, King Saud University, P.O. Box 2457, Riyadh 11451, Saudi Arabia; esali@ksu.edu.sa (E.E.); aamr@ksu.edu.sa (A.E.-G.E.A.); mehizia@ksu.edu.sa (A.A.A.); 2Bioavailability Center, National Organization for Drug Control and Research (NODCAR), Giza P.O. Box 29, Egypt; 3Clinical Pharmacy Department, College of Pharmacy, King Saud University, P.O. Box 2457, Riyadh 11451, Saudi Arabia; yasiri@ksu.edu.sa; 4Applied Organic Chemistry Department, National Research Center, Dokki, Cairo 12622, Egypt

**Keywords:** sofosbuvir, ledipasvir, daclatasvir, simeprevir, HPLC, hepatitis C

## Abstract

A simple, rapid, sensitive, and precise reversed-phase liquid chromatographic method was developed and validated for the simultaneous determination of four direct-acting antivirals, sofosbuvir (SF), ledipasvir (LD), declatasvir (DC), and simeprevir (SM), in their respective pharmaceutical formulations. Effective chromatographic separation was achieved on an Agilent Eclipse plus C8 column (250 mm × 4.6 mm, 5 µm) at 40 °C with gradient elution using a mobile phase composed of acetonitrile:phosphate buffer (pH 6.5). The quantification of SF and DC was based on peak area measurements at 260 nm, while the quantification of LD and SM was achieved at 330 nm. The linearity was acceptable from 1.0 to 20.0 μg/mL for the studied drugs, with correlation coefficients >0.999. The analytical performance of the newly proposed HPLC procedure was thoroughly validated according to ICH guidelines in terms of linearity, precision (RSD%, 0.39–1.57), accuracy (98.05–101.90%), specificity, limit of detection (LOD) (0.022–0.039 μg/mL), limit of quantification (LOQ) (0.067–0.118 μg/mL), and robustness. The validated HPLC method was successfully used to analyze the abovementioned drugs in their pure and dosage forms without interference from common excipients present in commercial formulations.

## 1. Introduction

Hepatitis C virus (HCV) infection is a worldwide health threat with 170–180 million infected individuals [1,2]. The virus can cause acute and chronic hepatitis infections, ranging in severity from a mild to serious lifelong illness [3]. Egypt has the largest epidemic of HCV worldwide, and according to the “Egyptian Demographic Health Survey (EDHS)”, approximately 13.2 out 90 million (14.7% of the population) suffer from HCV infection [4]. The early diagnosis of HCV infection is rare and the disease may go unnoticed until serious liver damage has occurred [5].

Until 2011, the treatment for HCV infection involved a combination of ribavirin and PEGylated-interferon. Improved knowledge of the HCV life cycle and its enzymes has facilitated the development of novel antiviral drugs. Recently, HCV treatment has been accomplished using direct-acting antivirals (DAAs), which are safer, more effective, and well-tolerated compared with older therapies and include sofosbuvir (SF), ledipasvir (LD), daclatasvir (DC), and simeprevir (SM).

SF is a prodrug of 2′-deoxy-2′-fluoro-2′-C-methyluridine monophosphate that is phosphorylated intracellularly to the active triphosphate form (Figure 1A). SF is a white to off-white crystalline solid having a molecular weight of 529.458 g/mol; has two pKa values: pKa_1_ = 9.38 (amide), pKa_2_ = 10.30; and is commonly used for the treatment of chronic HCV [6].

LD (molecular weight = 889.01 g/mol) has pKa value pKa_1_ = 11.33 and is a novel HCV NS5A inhibitor that has shown potent antiviral activity (Figure 1B). LD affects the HCV NS5A protein, which is involved in both RNA replication and aggregation of HCV virions [7]. A fixed-dose combination of SF/LD is currently recommended for the treatment of patients infected with the genotype 1HCV [8].

DC (molecular weight = 738.89 g/mol) is a white to yellow crystalline nonhygroscopic powder. It is freely soluble in water, dimethyl sulfoxide, and methanol; soluble in ethanol (95%); and practically insoluble in dichloromethane, tetrahydrofuran, acetonitrile, acetone, and ethylacetate. It has a LogP value of 4.67 and a pKa value of 11.15, and is a new direct-acting antiviral drug that targets different steps of the HCV life cycle (Figure 1C) [9]. DC binds to NS5A, an HCV protein involved in both viral RNA replication and virus particle aggregation, and inhibits its function [10]. The combination of DC with SF provides a powerful tool in the treatment of HCV genotype 3 [11].

SM (molecular weight = 749.94 g/mol) has two pKa values: pKa1 = 3.77, pKa2 = 1.61, and inhibits HCV replication by binding to HCV NS3/4A protease and inhibiting its activity (Figure 1D). HCV NS3/4A is necessary for viral replication [12,13] and SM/SF combination therapy has demonstrated efficacy and safety for the treatment of post-transplant HCV genotype 1 patients [14].

SF content was determined in its pure form [15], in tablet dosage form [16], or in the presence of its degradation products under various stress conditions [17,18] by reversed-phase high-performance liquid chromatography (RP-HPLC). The quantification of SF alone or with its metabolite in human plasma has also been achieved by liquid chromatography-tandem mass spectrometry (LC-MS/MS) [19,20,21]. 

LD content has been determined by UV spectrophotometry [22], RP-HPLC [23], and in rat plasma by ultra-performance liquid chromatography-tandem mass spectrometry (UPLC/MS/MS) [24]. Four HPLC methods for the estimation of DC in pharmaceutical formulations and bulk powder [25,26,27,28], and the LC-MS/MS method [29] for its quantitation in human plasma have been reported. Some methods have been published for SM determination in the presence of its degradation products [30] and in human plasma [31,32].

Several methods for the simultaneous determination of SF either with LD [19,33,34,35,36,37,38] or with SM have been published [13]. A single method has been published for the simultaneous determination of SF, LD, DC, and SM in plasma [39]. However, no analytical methods have been reported for the simultaneous determination of these antiviral drugs in their respective pharmaceutical dosage forms.

This study was performed to develop a simple, sensitive, rapid, and precise RP-HPLC method for determination of common direct-acting antivirals, SF, LD, DC, and SM, in bulk powder and pharmaceutical dosage forms simultaneously. The determination of these drugs in the same chromatographic run is important for quality control laboratories as it has the potential to save time and solvents.

## 2. Results and Discussion

As no method for the simultaneous determination of SF, LD, DC, and SM has been reported to date, the goal of this work was to develop a rapid and simple RP-HPLC method with gradient elution for their simultaneous determination in their pharmaceutical formulations.

The studied drugs were separated using the proposed method with good resolution and a suitable run time (8 min). Different parameters were tested to optimize the chromatographic separation of the studied drugs.

### 2.1. Method Development

To achieve optimum separation of the peaks of the studied drugs and improve the efficiency of the chromatographic system, different experimental parameters including column, column temperature, mobile phase composition, buffer pH, detection wavelength, and diluents were optimized.

#### 2.1.1. Choice of Column and Temperature

Different columns with different lengths, particle sizes, and internal diameters were tested at different temperatures, like the Nova pack C18 column (150 × 3.9 mm, 5 µm), BDS Hypersil, C18 column (250 × 4.6 mm, 5 µm), discovery cyano column C18 (150 × 4.6 mm, 5 µm), and Agilent Eclipse plus C8 column (250 × 4.6 mm, 5 µm). Symmetrical peaks with good resolution eluted within a reasonable time were obtained using an Agilent Eclipse plus C8 column (250 × 4.6 mm, 5 µm) at 40 °C. 

#### 2.1.2. Mobile Phase Composition and pH of Buffer 

Acetonitrile and methanol were tested as organic modifiers. Acetonitrile was selected because it provides improved separation of the studied drugs with a reasonable run time. Water, phosphate buffer, and acetate buffer were mixed with acetonitrile in different proportions, and 0.02 M anhydrous KH_2_PO_4_ buffer yielded the best separation. The effect of buffer pH was studied from pH 5 to 7, and pH 6.5 was selected as optimal because it provided the best resolution with symmetrical peaks. 

Different gradient elution modes were applied to improve separation of the studied drugs with reasonable retention times (<7 min) and good resolution between peaks (Figure 2). The selected gradient elution mode is described in Table 1.

#### 2.1.3. Choice of Detection Wavelength

The proper detection wavelength is important for method sensitivity. For good quantitation of the cited drugs, UV detection at 260 nm for SF and DC and 330 nm for LD and SM was selected based on the maximum absorption peak for each drug (Table 1). Daclatasvir and sofosbuvir were simultaneously separated with retention times of 2.262 and 2.867 min, respectively, and the time of difference between two peaks lasted less than a minute. The wavelength of the maximum absorption band for sofosbuvir was 260 nm and, for daclatasvir, it was 319 nm, with no detection for sofosbuvir. Short separation time did not allow enough time to change the wavelength to achieve maximum absorbance for both analytes individually. Therefore, the optimum wavelength to detect both compounds simultaneously was found to be 260 mm. 

#### 2.1.4. Choice of Diluents

The analytes were first dissolved in methanol as it allowed for proper dissolution of the analyzed drugs. Aliquots withdrawn were then brought to the specified volume with a solvent composed of acetonitrile and pH 6.5 phosphate buffer (50:50, *v*/*v*), providing good peak symmetry and sharpness to improve the resolution between the drugs.

### 2.2. Method Validation 

The method validation parameters, including system suitability, specificity, linearity, precision, accuracy, limit of detection (LOD), limit of quantification (LOQ), and robustness were optimized according to ICH guidelines for validation of analytical procedures [40].

#### 2.2.1. System Suitability

The system suitability test is an integral part of LC methods and is used to verify the adequacy of the system for analysis [41]. Assessment of column efficiency by number of theoretical plates (N), selectivity factor (α), retention factor (K), peak resolution (R), and peak tailing (T) indicated that the peaks were symmetrical and generally well-resolved (Table 2).

#### 2.2.2. Specificity

The specificity of an analytical method is defined as the ability to accurately determine the analyte in the presence of additional components, including impurities, degradation products, and excipients. The proposed method was tested for its specificity by assessing any changes resulting from the presence of common tablet additives, including lactose, starch, magnesium stearate, and talc. No interference occurred in the presence of these excipients, as indicated by the high mean percentage recovery and low SD.

In addition, the method specificity was evaluated using the peak purity plot for SF, LD, DC, and SM obtained by the photodiode array detector and peak purity information, which confirmed the purity of the drugs as the peak purity angle was smaller than the peak purity threshold (Figure 3).

#### 2.2.3. Linearity and Range

In this study, linearity was evaluated by the preparation of six different standard solutions containing the four drugs simultaneously at concentrations ranging from 1.0 to 20.0 µg/mL. Each concentration was analyzed three times and the calibration curves were obtained by plotting the peak area against the corresponding concentration. The linearity was determined by estimation of the regression line by the least squares method. Good calibration curve linearity was verified by the high correlation coefficients. The analytical data related to the calibration curves of the studied drugs are summarized in Table 3.

#### 2.2.4. Accuracy

The accuracy of the analytical method indicates the closeness of the assayed value to the theoretical value. In the newly developed method, the accuracy was verified by determination of SF, LD, DC, and SM in their pure forms at three different concentrations in triplicate for each concentration. The accuracy was also assessed by the standard addition technique to the commercial dosage forms containing SF and LD (Harvoni^®^ tablets, Gilead Sciences, Foster City, CA, USA), DC (Daclavirocyrl^®^ tablets, Marycl, Cairo, Egypt), and SM (Olysio^®^ capsules, Janssen-Cilag, Beerse, Belgium). Good recoveries were obtained in terms of accuracy and precision, indicating that the tablet excipients did not interfere in the quantification of the drugs in their dosage forms (Table 3).

#### 2.2.5. Precision

Precision is a measure of the closeness of the measurements to each other and was evaluated by repeatability (intra-day) and intermediate precision (inter-day). Precision judges the quality of the developed method. To evaluate precision, three different concentrations of SF, LD, DC, and SM (2.0, 8.0, and 16.0 µg/mL) were assayed in triplicate daily (repeatability) using the newly proposed method. The intermediate precision (inter-day) was determined by repeating the assay for the same concentrations on three successive days. Good RSD% values were obtained, indicating satisfactory precision of the proposed method (Table 3).

#### 2.2.6. Detection and Quantitation Limits

The limit of detection (LOD) is the minimum analyte concentration that can be detected. It was calculated as LOQ = 3.3 × σ/S, where σ is the standard deviation of the response and S is the calibration curve slope.

The limit of quantification (LOQ) is the minimum concentration that can be determined, under which the calibration plot is nonlinear, according to the ICH guidelines. It was calculated as LOD = 10 × σ/S.

The LOD values obtained by the proposed method were 0.03, 0.04, 0.02, and 0.03 µg/mL, and the LOQ values were 0.10, 0.12, 0.07, and 0.09 µg/mL for SF, LD, DC, and SM, respectively. These values indicate good sensitivity of the proposed method and its capability to detect and quantify the investigated drugs over a wide linear range.

#### 2.2.7. Robustness

Robustness indicates the ability of an analytical method to remain unaffected by small but deliberate changes to the chromatographic conditions and is an indication of the method reliability. Robustness was evaluated by changing the flow rate (2 mL·min^−1^ ± 0.1) and pH of the buffered solution in the mobile phase (pH 6.5 ± 0.1). No significant changes in the retention times were observed when the flow rate, pH, temperature, and buffer concentration were changed slightly. The RSD values indicated that the proposed method was sufficiently robust (Table 4).

### 2.3. Application of the Proposed Method for the Analysis of Commercial Tablets

The proposed and validated HPLC method was applied for the analysis of the studied drugs in their pharmaceutical dosage forms. The results presented in Table 5 are satisfactory, with good agreement with the labeled claim for each drug. There was no interference from the excipients present in the dosage form.

The validity of the proposed method was further assessed by performing the standard addition technique. The combined pre-analyzed tablet powders were spiked with SF, LD, DC, and SM pure standards, each at three different concentrations. The recovery of each pure drug was determined to be quantitative, as shown in Table 5.

Statistical comparisons between the results of the proposed method on pharmaceutical dosage forms and those of reported methods [27,31,38] were carried out using Student’s *t*-test for accuracy and *F*-test for precision. As shown in Table 6, the calculated *t*-test and *F*-values did not exceed the tabulated one at the 95% confidence level, revealing that there was no significant difference between the proposed and reported methods with respect to accuracy and precision.

## 3. Materials and Methods

Harvoni^®^ tablets (Batch no. VCKSDI), labeled to contain 400 and 90 mg of SF and LD per tablet, respectively, were obtained from Patheon, Inc. of Mississauga, Ontario for Gilead Sciences, Inc., Canada. Daclavirocyrl tablets (Batch no. 1631490), labeled to contain 60 mg DC per tablet, were obtained from Marcyrl Pharmaceutical Industries (Cairo, Egypt). Olysio^®^ capsules Janssen-Cilag (Beerse, Belgium), (Batch No. FJ 4H00) were obtained from Janssen-Cilag, labeled to contain 150 mg SM per tablet.

All chemicals used herein were of analytical grade and solvents were of HPLC grade. Methanol and acetonitrile were supplied by E-Merck (Darmstadt, Germany). Potassium dihydrogen orthophosphate (KH_2_PO_4_) and sodium hydroxide (NaOH) were from El Naser Pharmaceutical Chemicals Company (Egypt). Bi-distilled water was produced in-house (Aquatron Water Still, A4000D, England, UK) and membrane filters (0.45 μm Whatman) were used for mobile phase filtration.

Chromatographic separation was achieved using an Agilent Eclipse plus C8 column (250 mm × 4.6 mm, 5 µm) with a gradient elution based on a mobile phase consisting of KH_2_PO_4_ (pH adjusted to 6.5 with NaOH) and acetonitrile mixed in the ratios listed in Table 1. The flow rate was set to 2 mL/min and the column temperature was held at 40 °C. UV detection was programmed at 260 nm for SF and DC and 330 nm for LD and SM. The injection volume was 25 µL and data acquisition was performed using Agilent LC Chemstation software, version 32.

Standard stock solutions of SF, LD, DC, and SM (200.0 µg/mL) were prepared separately by dissolving 20 mg of each drug in 100 mL of methanol. Further dilutions were performed using a diluent (mixture of pH 6 phosphate buffer and acetonitrile 50:50 *v*/*v*) for preparation of the calibration standards.

Different mixed standard solutions of SF, LD, DC, and SM in the concentration range 1.0–20.0 µg/mL were prepared by transferring different aliquots of the standard stock solutions of each drug into the diluent in a volumetric flask (total volume of 10 mL). Then, 25 µL of each mixed standard solution was injected into the HPLC. The calibration curves for each drug were constructed by plotting the average peak areas against the concentration, and the corresponding regression equations were derived.

An accurately weighed amount equivalent to 400.0, 90.0, 60.0, and 150.0 mg of SF, LD, DC, and SM, respectively, was transferred from the respective pharmaceutical formulation to a 100 mL volumetric flask, dissolved in methanol by sonication for 15 min, and subsequently brought to the desired volume with methanol. The mixture was then filtered through a 0.45 μm syringe filter. From this solution, 1 mL was pipetted into a 50 mL volumetric flask and diluted to the specified mark with methanol. Aliquots containing suitable concentrations of the studied drugs that were diluted with diluent were analyzed as described under the “construction of calibration curves.” The quantitation of each drug was performed using the corresponding regression equation.

## 4. Conclusions

The proposed HPLC method with UV detection achieved the simple, sensitive, accurate, and rapid simultaneous determination of the SF, LD, DC, and SM HCV drugs. The developed method was applied for assaying the studied drugs in bulk and pharmaceutical formulations, and validated according to ICH guidelines. The novel method is highly sensitive (LODs 0.03, 0.04, 0.02, and 0.03 µg/mL for SF, LD, DC, and SM, respectively) and will be of great importance in quality control laboratories because it can save time (retention time < 7 min) and solvents by using the same mobile phase for the analysis of dosage and pure forms of the drugs simultaneously.

## Figures and Tables

**Figure 1 molecules-25-04611-f001:**
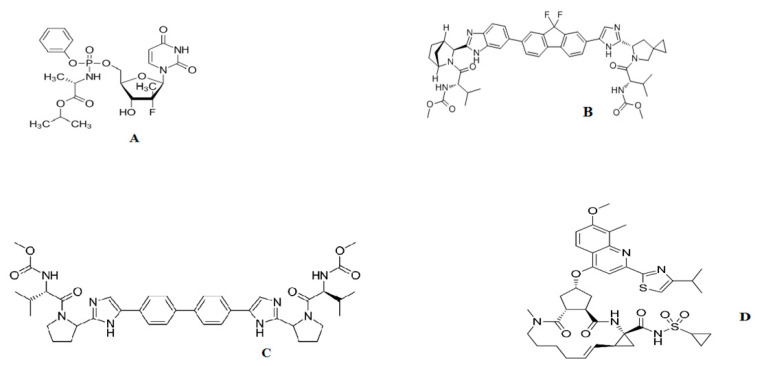
Chemical structures of (**A**) sofosbuvir, (**B**) ledipasvir, (**C**) daclatasvir, and (**D**) simeprevir.

**Figure 2 molecules-25-04611-f002:**
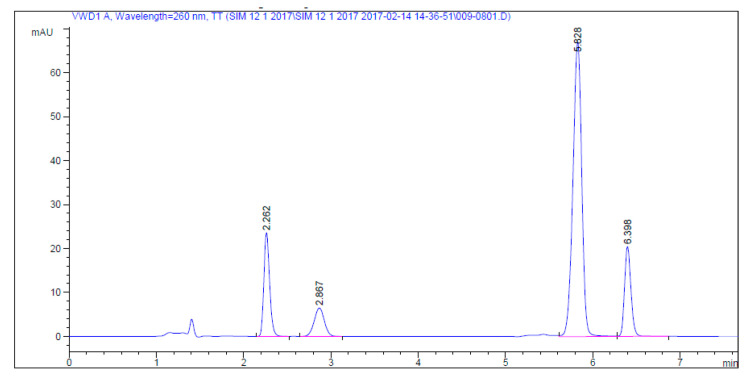
HPLC chromatogram of 2.0 µg/mL for each of daclatasvir (2.262 min), sofosbuvir (2.867 min), ledipasvir (5.828 min), and simeprevir (6.398 min) under the described chromatographic conditions.

**Figure 3 molecules-25-04611-f003:**
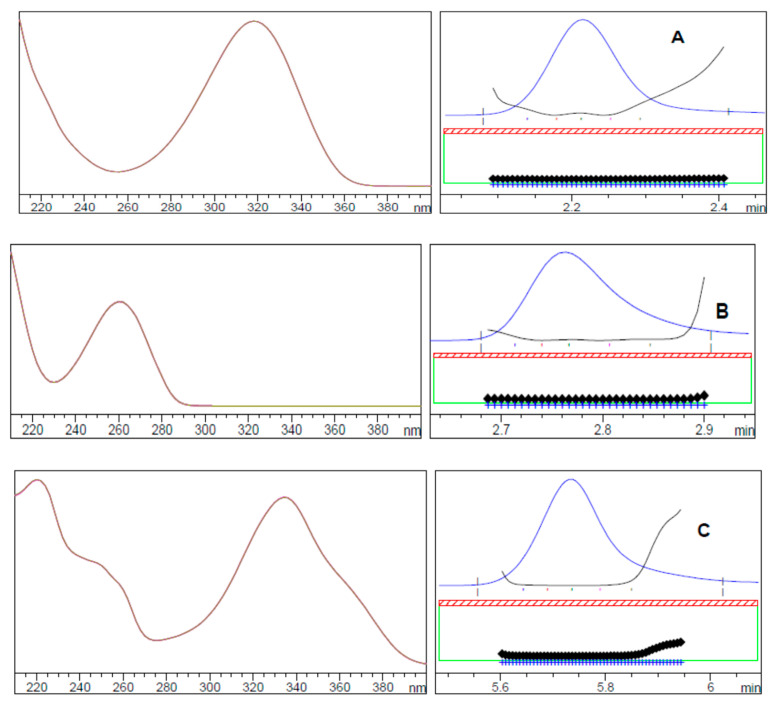
UV spectrum and peak purity curves of (**A**) daclatasvir, (**B**) sofosbuvir, (**C**) ledipasvir, and (**D**) simeprevir determined by a photodiode array (PDA) detector.

**Table 1 molecules-25-04611-t001:** Timetable of the validated gradient method.

Time (min)	Acetonitrile (%)	Buffer (%)	Wavelength (nm)
0.0	47	53	260
3.5	47	53	260
3.6	70	30	330
9.5	70	30	330
9.7	47	53	260
10.0	47	53	260

**Table 2 molecules-25-04611-t002:** System suitability tests for the developed HPLC method for the determination of sofosbuvir (SF), ledipasvir (LD), declatasvir (DC), and simeprevir (SM) (2.0 µg/mL).

Parameters	DC	SF		LD		SM	Reference Value [41]
	Obtained Value						
* Resolution (R)	2.73		12.57		4.46		R > 1.5
Tailing factor (T)	0.81	1.13		1.63		0.87	<1.5–2 (T = 1 for a typical symmetric peak)
Capacity factor (K)	1.13	1.6		4.49		5.29	1.0–10.0
** Selectivity factor (α)	1.22		2.21		1.14		α > 1.5
Number of theoretical plates (N)	9546	1951		6651		37350	More than 2000
Height Equivalent to the Theoretical Plate (HETP)	0.03	0.13		0.04		0.01	Smaller values indicate higher column efficiencies

* Resolution is difference in retention times between the two peaks. ** Selectivity factors is the distance between two chromatographic peaks.

**Table 3 molecules-25-04611-t003:** Regression and analytical parameters of the proposed HPLC method for the determination of SF, LD, DC, and SM.

Parameter	Drugs			
	SF	LD	DC	SM
Accuracy (Mean ± SD)	98.05 ± 0.61	101.90 ± 0.31	99.78 ± 1.21	100.87 ± 0.9
Precision:				
Repeatability ^a^	0.39	1.25	0.42	0.77
Intermediate precision ^b^	1.32	1.57	0.70	1.45
Linearity:				
Slope	4.20	37.784	13.768	16.589
Intercept	0.081	−0.6166	−0.597	0.0124
Correlation coefficient (r)	0.9995	0.9993	0.9998	0.9996
Range (µg/mL)	1.0–20.0			
LOD (µg/mL) ^c^	0.034	0.039	0.022	0.030
LOQ (µg/mL) ^d^	0.103	0.118	0.067	0.092

^a^ Average of three determinations of three concentrations of SF, LD, DC, and SM (2.0, 8.0, and 16.0 µg/mL) on the same day. ^b^ Average of three determinations of three concentrations of SF, LD, DC, and SM (2.0, 8.0 and 16.0 µg/mL) on three successive days. ^c^ LOD = 3.3·SD of the response/slope. ^d^ LOQ = 10·SD of the response/slope.

**Table 4 molecules-25-04611-t004:** Results of robustness study.

Drug	Chromatographic Condition					
	Mobile Phase Flow Rate mL/min					
	1.9		2.0		2.1	
	Retention Time (min) ± SD	RSD%	Retention Time (min) ± SD	RSD%	Retention Time (min) ± SD	RSD%
DC	2.452 ± 0.012	0.469	2.254 ± 0.007	0.302	2.035 ± 0.010	0.474
SF	2.983 ± 0.005	0.177	2.854 ± 0.012	0.428	2.605 ±0.006	0.225
LD	6.007 ± 0.008	0.136	5.828 ± 0.003	0.043	5.652 ± 0.002	0.035
SM	6.561 ± 0.033	0.51	6.393 ± 0.005	0.071	6.102 ± 0.001	0.016
	pH					
	6.4		6.5		6.6	
DC	2.257 ± 0.005	0.209	2.254 ± 0.007	0.302	2.233 ± 0.003	0.137
SF	2.862 ± 0.003	0.107	2.854 ± 0.012	0.428	2.861 ± 0.002	0.070
LD	5.829 ± 0.002	0.026	5.828 ± 0.003	0.043	5.831 ± 0.005	0.079
SM	6.393 ± 0.004	0.068	6.393 ± 0.005	0.071	6.399 ± 0.002	0.033
	Temperature					
	39 °C		40 °C		41 °C	
DC	2.259 ± 0.006	0.246	2.254 ± 0.007	0.302	2.255 ± 0.004	0.185
SF	2.883 ± 0.006	0.191	2.854 ± 0.012	0.428	2.829 ± 0.016	0.575
LD	5.832 ± 0.021	0.367	5.828 ± 0.003	0.043	5.852 ± 0.002	0.034
SM	6.390 ± 0.002	0.03	6.393 ± 0.005	0.071	6.391 ± 0.003	0.047
	Buffer Molarity					
	0.018 M		0.02 M		0.022 M	
DC	2.257 ± 0.008	0.355	2.254 ± 0.007	0.302	2.265 ± 0.010	0.444
SF	2.886 ± 0.019	0.666	2.854 ± 0.012	0.428	2.858 ± 0.024	0.855
LD	5.798 ± 0.023	0.397	5.828 ± 0.003	0.043	5.841 ± 0.034	0.581
SM	6.402 ± 0.023	0.364	6.393 ± 0.005	0.071	6.386 ± 0.005	0.072

**Table 5 molecules-25-04611-t005:** Determination of SF, LD, DC, and SM in their pharmaceutical formulations by the proposed HPLC method and application of standard addition technique.

Dosage Form	Component in the Product	Taken (µg/mL)	Found % ^a^ ± SD	Pure Added (µg/mL)	Recovery % ^a^	Mean ± SD
Harvoni tablet (400 mg SF and 90 mg LD) B.N. VCKSDI	SF (400 mg)	8.0	98.37 ± 1.51	6.0	97.73	97.44 ± 0.26
				8.0	97.38	
				10.0	97.22	
	LD (90 mg)	1.8	101.00 ± 0.70	1.2	99.01	99.27 ± 0.98
				1.8	100.36	
				2.0	98.44	
Daclavirocyrl B.N. 1631490	DC (200 mg)	1.2	100.81 ± 1.16	1.0	98.25	98.31 ± 0.45
				1.20	97.89	
				2.0	98.78	
Olysio^®^ B.N. FJ4H00	SM (150 mg)	3.0	101.19 ± 0.60	2.0	99.06	100.12 ± 0.99
				3.0	100.27	
				4.0	101.02	

^a^ Average of three determinations.

**Table 6 molecules-25-04611-t006:** Statistical comparison between results of the dosage forms of the proposed and those of the reported methods.

Items	The Proposed HPLC Method				Reported Methods			
	SF	LD	DC	SM	SF ^a^ [38]	LD ^a^ [38]	DC ^b^ [27]	SM ^c^ [31]
**Mean ± SD * %**	98.37 ± 1.51	101.00 ± 0.70	100.81 ± 1.16	101.19 ± 0.60	99.19 ± 1.02	101.26 ± 0.79	100.54 ± 0.97	100.67 ± 0.951
**Variance**	2.280	0.490	1.346	0.360	1.040	0.624	0.941	0.904
**N**	6	6	6	6	6	6	5	5
**Student’s *t* test**	1.102 (2.228) **	0.603 (2.228) **	0.372 (2.262) **	1.107 (2.262) **				
***F* value**	2.19 (5.05) **	1.27 (5.05) **	1.43 (6.26) **	2.511 (5.19) **				

* Average of three determinations. ** The values between parentheses correspond to the theoretical values of *t* and *F* (*p* = 0.05). ^a^ Reported method, ratio subtraction spectrophotometric method for simultaneous determination of SF at 261 nm and LD at zero-order spectra at 333 nm. ^b^ Reported method, HPLC method for determination of DC using acetonitrile:methanol (70:30 *v*/*v*) and UV detection at 230 nm. ^c^ Reported method, HPLC method for determination of SM using phosphate buffer (pH 6, 52.5 mM):acetonitrile (30:70 *v*/*v*) as the mobile phase and UV detection at 225 nm.

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
