# Peer review of "Validated Reversed-Phase Liquid Chromatographic Method with Gradient Elution for Simultaneous Determination of the Antiviral Agents: Sofosbuvir, Ledipasvir, Daclatasvir, and Simeprevir in Their Dosage Forms"

_molecules, 2020, doi:10.3390/molecules25204611_

Round 1
Reviewer 1 Report
The manuscript reported the development of HPLC method for simultaneous determination of antiviral drugs. Presented method is applicable for analysis of pharmaceutical preparatives, but the paper is not suitable for publication in Molecules in this form. The following remarks were noted:
1) Pages 1-3, Introduction: Initially, information on analytical methods used to determine target substances (method parameters, analyzed samples, sample preparation, ...) could be expanded.
2) Page 1, Line 26: The value of correlation coefficient listed in abstract (>0.9999) is not consistent with the results listed in table 3 (0.9993-0.9998).
3) Page 1, Abstract: The main results of the analytical parameters of the HPLC method obtained should be reported in the abstract.
4) Page 3, Line 59: I assume the authors thought "Sofosbuvir/SF content was determined in …” instead of "SF content was measured in …”
5) Pages 3-5, part Method development: The whole part is very brief. It would be appropriate to comment in more detail on the tested parameters of the method, to discuss the optimized parameters, to supplement the results of individual tested conditions (achieved k, R, alpha, peaks symmetry, …).
6) Page 5, line 129: The authors state "selectivity factor (alpha)” (Line 129), as well as “Relative retention (alpha)” (Table 2). Please unify the parameter name.
7) Page 5, Table 2: The concentration of the reference solution used for testing should be reported in Table 2 legend.
8) Page 5, lines 137-138: Lactose, starch, magnesium stearate, talc were used for selectivity study. Did the authors expect the possibility of interference of these substances during separation? Some substances absorb radiation in another region of the UV spectrum and thus the possibility of interference is minimal. Please explain the reason for the selection of substances for the selective study.
9) Page 6, Figure 3: Edit the text for Figure 3 and include "UV spectrum of analytes" to the text.
10) Page 6, Figure 3: From the UV spectrum of declatasvir, the maximum of the absorption band at a wavelength of 319 nm is probably. Justify your choice - 260 nm (page 5, line 115)!
11) Page 6, linearity and range: Indicate the number of repetitions of preparation of analyte solutions used to evaluate linearity and other validation parameters.
12) Page 8, Robustness: The small change in column temperature (+- 2 oC) should also be included in the robustness evaluation of the method.
13) Page 8, line 195: Table 4 is not included in the manuscript.
14) Page numbers are not included in References nr. 15, 18, 20 in references list.
In conclusion, I advise a major revision of manuscript concerning on addition of more detailed information and results in method development part. It would be interesting to point out the advantages of the developed analytical method in comparison to previously published methods.
Reviewer 2 Report
The manuscript is well written and the results are clearly presented, however, some concerns need to be addressed before this work can be accepted for publication in Molecules.
1- Line 63, cite recent publication " Estimation of sofosbuvir and its metabolite in human plasma using LC/MSMS , https://doi.org/10.3390/molecules24071302
2- The specifications of the tested columns i.e. particle size,length,stationary phase, etc.. should mentioned.
3- Since the selection of the pH of the mobile phase is PKa dependent, so a table showing the Pka's and some physicochemical parameters of the tested drugs is recommended.
4- Table 2, reference value of the number of theoretical plates which more than 2000 should be mentioned
5- Line 195 table 4 showing the robustness results is missing
Reviewer 3 Report
Review result: Minor changes ID: molecules-919934 Title: Validated reversed phase liquid chromatographic method with gradient elution for simultaneous determination of the antiviral agents sofosbuvir, ledipasvir, daclatasvir, and simeprevir in their dosage forms Authors: Essam Ezzeldin, Nisreen Abo-Talib, Marwa Ammam, Gamal Mostafa, Yousef Asiri Originality: fair Technical quality: good Clarity of presentation: fair Importance to field: good Summary: The paper presents the results of optimization studies for the determination of four direct-acting antivirals, sofosbuvir (SF), ledipasvir (LD), declatasvir (DC), and simeprevir (SM) by reversed-phase liquid chromatographic method. However, the paper needs minor improvement before being considered for publication. To support my decision (Publish after MINOR changes), I am providing the list of comments and suggested corrections, hoping this will help the authors improve and publish the article in the future. Title: The Title may be shorten: ‘Validated reversed phase liquid chromatographic method with gradient elution for simultaneous determination of the antiviral agents in their dosage forms’ or at least the title should have a colon after ‘… antiviral agents’ i.e. ‘Validated reversed phase liquid chromatographic method with gradient elution for simultaneous determination of the antiviral agents: sofosbuvir, ledipasvir, daclatasvir, and simeprevir in their dosage forms’. Affiliations: Affiliation number 4 (‘Micro-Analytical Laboratory, Applied Organic Chemistry Department, National Research Center, Dokki, 15 Cairo 12622, Egypt’) is not assigned to any authors. Please remove it or assign one of the authors to this location. Keywords: Please consider adding: analytical methods, Hepatitis C virus (HCV) Abstract: No comments Introduction: Please improve the visual aspects of the Figure 1. I suggest using one selected software for plotting and presenting all four structure. Results and Discussion: - This section consists only results. - There is not much discussion in this section. I would be great to present general discussion of the results. In current form the paper looks more like a ‘Communication’ not ‘Full Article’. - A comparison of the separation results with the separation achieved in other papers, - Describe the process of optimization study. How changes of different parameters (column, temperature, mobile phase composition, buffer pH, detection wavelength, and diluents) affected the separation. For example, in section 2.1.1. the authors wrote: ‘Different columns with different lengths, particle sizes, and internal diameters were tested at different temperatures’. The readers would like to know what columns, how many, what temperature? Then it would be great to show at least two cases out of the above set (i.e. Figure 2 and other intermediate case). - The above comments is valid for entire 2.1 section of the ‘Results and Discussion’. - Please improve quality of Figure 2 – now looks like copy/paste of the print screen. - Table 2: Please add reference for the ‘Reference value’ column. - Figure 3: Please discuss what is on the left and right side of the figure. What do the "diamond" and "cross" -shaped profiles on the right side panel of this figure represent? Materials and Methods: No comments References: No comments
Round 2
Reviewer 1 Report
Authors includes many comments into the text. They reworked manuscript in the relevant sections, however comment focused on choice of detection wavelength was not commented:
- From the UV spectrum of declatasvir (Fig 3A) is evident that the wavelength of maximum of absorption band is 319 nm. This detection wavelength is suitable from the aspect of method sensitivity and also selectivity. However, the authors chose the wavelength of 260 nm (Table 1; chapter 2.1.3-Choice of detection wavelength). The choice of this parameter must be explained in the relevant text.
Minor comments:
- Table 4: Include the unit for “Retention time” - (min) in the table.
- References nr. 15, 18, 20 are incorrectly cited:
15. …… Asian Journal of Pharmaceutical and Clinical Research, 9(9), 61-66.
18. …… Journal of chromatographic science 54 (9), 1631-1640.
20. …….International Journal of Pharmacy and Pharmaceutical Sciences 9(3), 35-41.
Author Response
Response to reviewer 1 comment
Authors includes many comments into the text. They reworked manuscript in the relevant sections, however comment focused on choice of detection wavelength was not commented:
- From the UV spectrum of declatasvir (Fig 3A) is evident that the wavelength of maximum of absorption band is 319 nm. This detection wavelength is suitable from the aspect of method sensitivity and also selectivity. However, the authors chose the wavelength of 260 nm (Table 1; chapter 2.1.3-Choice of detection wavelength). The choice of this parameter must be explained in the relevant text.
The reason for the choice of detection of wave length has been added to manuscript (Page 4 Lines 121-127).
(Daclatasvir and sofosbuvir were simultaneously separated with the retention time of 2.262 and 2.867 minutes, respectively and the time of difference between two peaks lasts less than a minute. The wavelength of maximum absorption band for sofosbuvir was 260 nm and for declatasvir was 319 nm with no detection for sofosbuvir. Short separation time didn’t allow enough time to change the wave length to achieve the maximum absorbance for both analytes individually. Therefor the optimum wave length to detect both compounds simultaneously was found to be 260mm.
)
Table 4: Include the unit for “Retention time” - (min) in the table
The time unit has been added.
- References nr. 15, 18, 20 are incorrectly cited:
References has been revised
Reviewer 2 Report
The manuscript has improved significantly, and I recommend the publication of the revised version.
Author Response
Thank you for your revision